# A Simple Unified Framework for Detecting Out-of-Distribution Samples and Adversarial Attacks

**Kimin Lee**[1], **Kibok Lee**[2], **Honglak Lee**[3,2], **Jinwoo Shin**[1,4]
[1]Korea Advanced Institute of Science and Technology (KAIST)
[2]University of Michigan
[3]Google Brain
[4]AItrics

## Abstract

Detecting test samples drawn sufficiently far away from the training distribution statistically or adversarially is a fundamental requirement for deploying a good classifier in many real-world machine learning applications. However, deep neural networks with the softmax classifier are known to produce highly overconfident posterior distributions even for such abnormal samples. In this paper, we propose a simple yet effective method for detecting any abnormal samples, which is applicable to any pre-trained softmax neural classifier. We obtain the class conditional Gaussian distributions with respect to (low- and upper-level) features of the deep models under Gaussian discriminant analysis, which result in a confidence score based on the Mahalanobis distance. While most prior methods have been evaluated for detecting either out-of-distribution or adversarial samples, but not both, the proposed method achieves the state-of-the-art performances for both cases in our experiments. Moreover, we found that our proposed method is more robust in harsh cases, e.g., when the training dataset has noisy labels or small number of samples. Finally, we show that the proposed method enjoys broader usage by applying it to class-incremental learning: whenever out-of-distribution samples are detected, our classification rule can incorporate new classes well without further training deep models.

## 1 Introduction

Deep neural networks (DNNs) have achieved high accuracy on many classification tasks, e.g., speech recognition [1], object detection [9] and image classification [12]. However, measuring the predictive uncertainty still remains a challenging problem [20, 21]. Obtaining well-calibrated predictive uncertainty is indispensable since it could be useful in many machine learning applications (e.g., active learning [8] and novelty detection [18]) as well as when deploying DNNs in real-world systems [2], e.g., self-driving cars and secure authentication system [6, 30].

The predictive uncertainty of DNNs is closely related to the problem of detecting abnormal samples that are drawn far away from in-distribution (i.e., distribution of training samples) statistically or adversarially. For detecting out-of-distribution (OOD) samples, recent works have utilized the confidence from the posterior distribution [13, 21]. For example, Hendrycks & Gimpel [13] proposed the maximum value of posterior distribution from the classifier as a baseline method, and it is improved by processing the input and output of DNNs [21]. For detecting adversarial samples, confidence scores were proposed based on density estimators to characterize them in feature spaces of DNNs [7]. More recently, Ma et al. [22] proposed the local intrinsic dimensionality (LID) and empirically showed that the characteristics of test samples can be estimated effectively using the

LID. However, most prior works on this line typically do not evaluate both OOD and adversarial samples. To best of our knowledge, no universal detector is known to work well on both tasks.

**Contribution.** In this paper, we propose a simple yet effective method, which is applicable to any pre-trained softmax neural classifier (without re-training) for detecting abnormal test samples including OOD and adversarial ones. Our high-level idea is to measure the probability density of test sample on feature spaces of DNNs utilizing the concept of a "generative" (distance-based) classifier. Specifically, we assume that pre-trained features can be fitted well by a class-conditional Gaussian distribution since its posterior distribution can be shown to be equivalent to the softmax classifier under Gaussian discriminant analysis (see Section 2.1 for our justification). Under this assumption, we define the confidence score using the Mahalanobis distance with respect to the closest class-conditional distribution, where its parameters are chosen as empirical class means and tied empirical covariance of training samples. To the contrary of conventional beliefs, we found that using the corresponding generative classifier does not sacrifice the softmax classification accuracy. Perhaps surprisingly, its confidence score outperforms softmax-based ones very strongly across multiple other tasks: detecting OOD samples, detecting adversarial samples and class-incremental learning.

We demonstrate the effectiveness of the proposed method using deep convolutional neural networks, such as DenseNet [14] and ResNet [12] trained for image classification tasks on various datasets including CIFAR [15], SVHN [28], ImageNet [5] and LSUN [32]. First, for the problem of detecting OOD samples, the proposed method outperforms the current state-of-the-art method, ODIN [21], in all tested cases. In particular, compared to ODIN, our method improves the true negative rate (TNR), i.e., the fraction of detected OOD (e.g., LSUN) samples, from $45.6\%$ to $90.9\%$ on ResNet when $95\%$ of in-distribution (e.g., CIFAR-100) samples are correctly detected. Next, for the problem of detecting adversarial samples, e.g., generated by four attack methods such as FGSM [10], BIM [16], DeepFool [26] and CW [3], our method outperforms the state-of-the-art detection measure, LID [22]. In particular, compared to LID, ours improves the TNR of CW from $82.9\%$ to $95.8\%$ on ResNet when $95\%$ of normal CIFAR-10 samples are correctly detected.

We also found that our proposed method is more robust in the choice of its hyperparameters as well as against extreme scenarios, e.g., when the training dataset has some noisy, random labels or a small number of data samples. In particular, Liang et al. [21] tune the hyperparameters of ODIN using validation sets of OOD samples, which is often impossible since the knowledge about OOD samples is not accessible a priori. We show that hyperparameters of the proposed method can be tuned only using in-distribution (training) samples, while maintaining its performance. We further show that the proposed method tuned on a simple attack, i.e., FGSM, can be used to detect other more complex attacks such as BIM, DeepFool and CW.

Finally, we apply our method to class-incremental learning [29]: new classes are added progressively to a pre-trained classifier. Since the new class samples are drawn from an out-of-training distribution, it is natural to expect that one can classify them using our proposed metric without re-training the deep models. Motivated by this, we present a simple method which accommodates a new class at any time by simply computing the class mean of the new class and updating the tied covariance of all classes. We show that the proposed method outperforms other baseline methods, such as Euclidean distance-based classifier and re-trained softmax classifier. This evidences that our approach have a potential to apply to many other related machine learning tasks, such as active learning [8], ensemble learning [19] and few-shot learning [31].

## 2 Mahalanobis distance-based score from generative classifier

Given deep neural networks (DNNs) with the softmax classifier, we propose a simple yet effective method for detecting abnormal samples such as out-of-distribution (OOD) and adversarial ones. We first present the proposed confidence score based on an induced generative classifier under Gaussian discriminant analysis (GDA), and then introduce additional techniques to improve its performance. We also discuss how the confidence score is applicable to incremental learning.

### 2.1 Why Mahalanobis distance-based score?

**Derivation of generative classifiers from softmax ones.** Let $\mathbf{x} \in \mathcal{X}$ be an input and $y \in \mathcal{Y} = \{1, \cdots, C\}$ be its label. Suppose that a pre-trained softmax neural classifier is given:

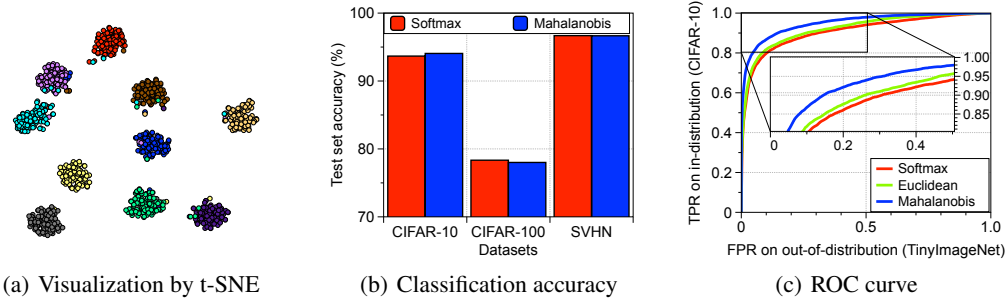

|  | (a) Visualization by t-SNE | (b) Classification accuracy | (c) ROC curve |

Figure 1: Experimental results under the ResNet with 34 layers. (a) Visualization of final features from ResNet trained on CIFAR-10 by t-SNE, where the colors of points indicate the classes of the corresponding objects. (b) Classification test set accuracy of ResNet on CIFAR-10, CIFAR-100 and SVHN datasets. (c) Receiver operating characteristic (ROC) curves: the x-axis and y-axis represent the false positive rate (FPR) and true positive rate (TPR), respectively.

$P(y = c | \mathbf{x}) = \frac{\exp(\mathbf{w}_c^\top f(\mathbf{x}) + b_c)}{\sum_{c'} \exp(\mathbf{w}_{c'}^\top f(\mathbf{x}) + b_{c'})}$, where $\mathbf{w}_c$ and $b_c$ are the weight and the bias of the softmax classifier for class $c$, and $f(\cdot)$ denotes the output of the penultimate layer of DNNs. Then, without any modification on the pre-trained softmax neural classifier, we obtain a generative classifier assuming that a class-conditional distribution follows the multivariate Gaussian distribution. Specifically, we define $C$ class-conditional Gaussian distributions with a tied covariance $\boldsymbol{\Sigma}$: $P(f(\mathbf{x}) | y = c) = \mathcal{N}(f(\mathbf{x}) | \mu_c, \boldsymbol{\Sigma})$, where $\mu_c$ is the mean of multivariate Gaussian distribution of class $c \in \{1, ..., C\}$. Here, our approach is based on a simple theoretical connection between GDA and the softmax classifier: the posterior distribution defined by the generative classifier under GDA with tied covariance assumption is equivalent to the softmax classifier (see the supplementary material for more details). Therefore, the pre-trained features of the softmax neural classifier $f(\mathbf{x})$ might also follow the class-conditional Gaussian distribution.

To estimate the parameters of the generative classifier from the pre-trained softmax neural classifier, we compute the empirical class mean and covariance of training samples $\{(\mathbf{x}_1, y_1), \ldots, (\mathbf{x}_N, y_N)\}$:

$$\widehat{\mu}_c = \frac{1}{N_c} \sum_{i:y_i=c} f(\mathbf{x}_i), \ \ \widehat{\boldsymbol{\Sigma}} = \frac{1}{N} \sum_c \sum_{i:y_i=c} (f(\mathbf{x}_i) - \widehat{\mu}_c)(f(\mathbf{x}_i) - \widehat{\mu}_c)^\top, \quad (1)$$

where $N_c$ is the number of training samples with label $c$. This is equivalent to fitting the class-conditional Gaussian distributions with a tied covariance to training samples under the maximum likelihood estimator.

**Mahalanobis distance-based confidence score.** Using the above induced class-conditional Gaussian distributions, we define the confidence score $M(\mathbf{x})$ using the Mahalanobis distance between test sample $\mathbf{x}$ and the closest class-conditional Gaussian distribution, i.e.,

$$M(\mathbf{x}) = \max_c \ - (f(\mathbf{x}) - \widehat{\mu}_c)^\top \widehat{\boldsymbol{\Sigma}}^{-1} (f(\mathbf{x}) - \widehat{\mu}_c). \quad (2)$$

Note that this metric corresponds to measuring the log of the probability densities of the test sample. Here, we remark that abnormal samples can be characterized better in the representation space of DNNs, rather than the "label-overfitted" output space of softmax-based posterior distribution used in the prior works [13, 21] for detecting them. It is because a confidence measure obtained from the posterior distribution can show high confidence even for abnormal samples that lie far away from the softmax decision boundary. Feinman et al. [7] and Ma et al. [22] process the DNN features for detecting adversarial samples in a sense, but do not utilize the Mahalanobis distance-based metric, i.e., they only utilize the Euclidean distance in their scores. In this paper, we show that Mahalanobis distance is significantly more effective than the Euclidean distance in various tasks.

**Experimental supports for generative classifiers.** To evaluate our hypothesis that trained features of DNNs support the assumption of GDA, we measure the classification accuracy as follows:

$$\widehat{y}(\mathbf{x}) = \arg\min_c (f(\mathbf{x}) - \widehat{\mu}_c)^\top \widehat{\boldsymbol{\Sigma}}^{-1} (f(\mathbf{x}) - \widehat{\mu}_c). \quad (3)$$

---

**Algorithm 1** Computing the Mahalanobis distance-based confidence score.

---

**Input:** Test sample $\mathbf{x}$, weights of logistic regression detector $\alpha_\ell$, noise $\varepsilon$ and parameters of Gaussian distributions $\{\widehat{\mu}_{\ell,c}, \widehat{\mathbf{\Sigma}}_\ell : \forall \ell, c\}$

---

Initialize score vectors: $\mathbf{M}(\mathbf{x}) = [M_\ell : \forall \ell]$
**for** each layer $\ell \in 1, \ldots, L$ **do**

    Find the closest class: $\widehat{c} = \arg\min_c \ (f_\ell(\mathbf{x}) - \widehat{\mu}_{\ell,c})^\top \widehat{\mathbf{\Sigma}}_\ell^{-1}(f_\ell(\mathbf{x}) - \widehat{\mu}_{\ell,c})$

    Add small noise to test sample: $\widehat{\mathbf{x}} = \mathbf{x} - \varepsilon\text{sign}\left(\bigtriangledown_{\mathbf{x}} (f_\ell(\mathbf{x}) - \widehat{\mu}_{\ell,\widehat{c}})^\top \widehat{\mathbf{\Sigma}}_\ell^{-1} (f_\ell(\mathbf{x}) - \widehat{\mu}_{\ell,\widehat{c}})\right)$

    Computing confidence score: $M_\ell = \max_c - (f_\ell(\widehat{\mathbf{x}}) - \widehat{\mu}_{\ell,c})^\top \widehat{\mathbf{\Sigma}}_\ell^{-1} (f_\ell(\widehat{\mathbf{x}}) - \widehat{\mu}_{\ell,c})$

**end for**
**return** Confidence score for test sample $\sum_\ell \alpha_\ell M_\ell$

---

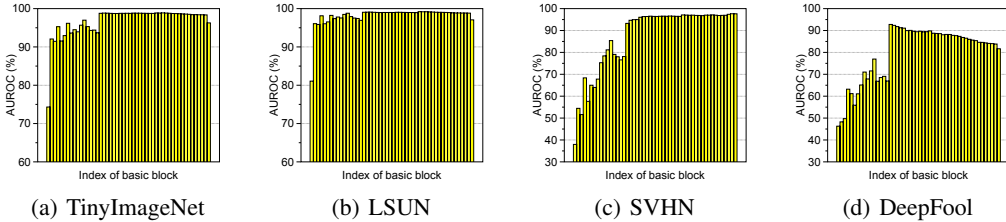

    (a) TinyImageNet      (b) LSUN      (c) SVHN      (d) DeepFool

Figure 2: AUROC (%) of threshold-based detector using the confidence score in (2) computed at different basic blocks of DenseNet trained on CIFAR-10 dataset. We measure the detection performance using (a) TinyImageNet, (b) LSUN, (c) SVHN and (d) adversarial (DeepFool) samples.

We remark that this corresponds to predicting a class label using the posterior distribution from generative classifier with the uniform class prior. Interestingly, we found that the softmax accuracy (red bar) is also achieved by the Mahalanobis distance-based classifier (blue bar), while conventional knowledge is that a generative classifier trained from scratch typically performs much worse than a discriminative classifier such as softmax. For visual interpretation, Figure 1(a) presents embeddings of final features from CIFAR-10 test samples constructed by t-SNE [23], where the colors of points indicate the classes of the corresponding objects. One can observe that all ten classes are clearly separated in the embedding space, which supports our intuition. In addition, we also show that Mahalanobis distance-based metric can be very useful in detecting out-of-distribution samples. For evaluation, we obtain the receiver operating characteristic (ROC) curve using a simple threshold-based detector by computing the confidence score $M(\mathbf{x})$ on a test sample $\mathbf{x}$ and decide it as positive (i.e., in-distribution) if $M(\mathbf{x})$ is above some threshold. The Euclidean distance, which only utilizes the empirical class means, is considered for comparison. We train ResNet on CIFAR-10, and TinyImageNet dataset [5] is used for an out-of-distribution. As shown in Figure 1(c), the Mahalanobis distance-based metric (blue bar) performs better than Euclidean one (green bar) and the maximum value of the softmax distribution (red bar).

## 2.2 Calibration techniques

**Input pre-processing.** To make in- and out-of-distribution samples more separable, we consider adding a small controlled noise to a test sample. Specifically, for each test sample $\mathbf{x}$, we calculate the pre-processed sample $\widehat{\mathbf{x}}$ by adding the small perturbations as follows:

$$\widehat{\mathbf{x}} = \mathbf{x} + \varepsilon\text{sign}\left(\bigtriangledown_{\mathbf{x}} M(\mathbf{x})\right) = \mathbf{x} - \varepsilon\text{sign}\left(\bigtriangledown_{\mathbf{x}} (f(\mathbf{x}) - \widehat{\mu}_{\widehat{c}})^\top \widehat{\mathbf{\Sigma}}^{-1} (f(\mathbf{x}) - \widehat{\mu}_{\widehat{c}})\right), \quad (4)$$

where $\varepsilon$ is a magnitude of noise and $\widehat{c}$ is the index of the closest class. Next, we measure the confidence score using the pre-processed sample. We remark that the noise is generated to increase the proposed confidence score (2) unlike adversarial attacks [10]. In our experiments, such perturbation can have stronger effect on separating the in- and out-of-distribution samples. We remark that similar input pre-processing was studied in [21], where the perturbations are added to increase the softmax score of the predicted label. However, our method is different in that the noise is generated to increase the proposed metric.

---
**Algorithm 2** Updating Mahalanobis distance-based classifier for class-incremental learning.

---
**Input:** set of samples from a new class $\{\mathbf{x}_i : \forall i = 1 \dots N_{C+1}\}$, mean and covariance of observed classes $\{\widehat{\mu}_c : \forall c = 1 \dots C\}, \widehat{\boldsymbol{\Sigma}}$

---
Compute the new class mean: $\widehat{\mu}_{C+1} \leftarrow \frac{1}{N_{C+1}} \sum_i f(\mathbf{x}_i)$

Compute the covariance of the new class: $\widehat{\boldsymbol{\Sigma}}_{C+1} \leftarrow \frac{1}{N_{C+1}} \sum_i (f(\mathbf{x}_i) - \widehat{\mu}_{C+1})(f(\mathbf{x}_i) - \widehat{\mu}_{C+1})^\top$

Update the shared covariance: $\widehat{\boldsymbol{\Sigma}} \leftarrow \frac{C}{C+1}\widehat{\boldsymbol{\Sigma}} + \frac{1}{C+1}\widehat{\boldsymbol{\Sigma}}_{C+1}$

**return** Mean and covariance of all classes $\{\widehat{\mu}_c : \forall c = 1 \dots C + 1\}, \widehat{\boldsymbol{\Sigma}}$

---

**Feature ensemble.** To further improve the performance, we consider measuring and combining the confidence scores from not only the final features but also the other low-level features in DNNs. Formally, given training data, we extract the $\ell$-th hidden features of DNNs, denoted by $f_\ell(\mathbf{x})$, and compute their empirical class means and tied covariances, i.e., $\widehat{\mu}_{\ell,c}$ and $\widehat{\boldsymbol{\Sigma}}_\ell$. Then, for each test sample $\mathbf{x}$, we measure the confidence score from the $\ell$-th layer using the formula in (2). One can expect that this simple but natural scheme can bring an extra gain in obtaining a better calibrated score by extracting more input-specific information from the low-level features. We measure the area under ROC (AUROC) curves of the threshold-based detector using the confidence score in (2) computed at different basic blocks of DenseNet [14] trained on CIFAR-10 dataset, where the overall trends on ResNet are similar. Figure 2 shows the performance on various OOD samples such as SVHN [28], LSUN [32], TinyImageNet and adversarial samples generated by DeepFool [26], where the dimensions of the intermediate features are reduced using average pooling (see Section 3 for more details). As shown in Figure 2, the confidence scores computed at low-level features often provide better calibrated ones compared to final features (e.g., LSUN, TinyImageNet and DeepFool). To further improve the performance, we design a feature ensemble method as described in Algorithm 1. We first extract the confidence scores from all layers, and then integrate them by weighted averaging: $\sum_\ell \alpha_\ell M_\ell(\mathbf{x})$, where $M_\ell(\cdot)$ and $\alpha_\ell$ is the confidence score at the $\ell$-th layer and its weight, respectively. In our experiments, following similar strategies in [22], we choose the weight of each layer $\alpha_\ell$ by training a logistic regression detector using validation samples. We remark that such weighted averaging of confidence scores can prevent the degradation on the overall performance even in the case when the confidence scores from some layers are not effective: the trained weights (using validation) would be nearly zero for those ineffective layers.

## 2.3 Class-incremental learning using Mahalanobis distance-based score

As a natural extension, we also show that the Mahalanobis distance-based confidence score can be utilized in class-incremental learning tasks [29]: a classifier pre-trained on base classes is progressively updated whenever a new class with corresponding samples occurs. This task is known to be challenging since one has to deal with catastrophic forgetting [24] with a limited memory. To this end, recent works have been toward developing new training methods which involve a generative model or data sampling, but adopting such training methods might incur expensive back-and-forth costs. Based on the proposed confidence score, we develop a simple classification method without the usage of complicated training methods. To do this, we first assume that the classifier is well pre-trained with a certain amount of base classes, where the assumption is quite reasonable in many practical scenarios.[1] In this case, one can expect that not only the classifier can detect OOD samples well, but also might be good for discriminating new classes, as the representation learned with the base classes can characterize new ones. Motivated by this, we present a Mahalanobis distance-based classifier based on (3), which tries to accommodate a new class by simply computing and updating the class mean and covariance, as described in Algorithm 2. The class-incremental adaptation of our confidence score shows its potential to be applied to a wide range of new applications in the future.

| Method | Feature ensemble | Input pre-processing | TNR at TPR 95% | AUROC | Detection accuracy | AUPR in | AUPR out |
|---|---|---|---|---|---|---|---|
| Baseline [13] | - | - | 32.47 | 89.88 | 85.06 | 85.40 | 93.96 |
| ODIN [21] | - | - | 86.55 | 96.65 | 91.08 | 92.54 | 98.52 |
| Mahalanobis (ours) | - | - | 54.51 | 93.92 | 89.13 | 91.56 | 95.95 |
| | - | ✓ | 92.26 | 98.30 | 93.72 | 96.01 | 99.28 |
| | ✓ | - | 91.45 | 98.37 | 93.55 | 96.43 | 99.35 |
| | ✓ | ✓ | **96.42** | **99.14** | **95.75** | **98.26** | **99.60** |

Table 1: Contribution of each proposed method on distinguishing in- and out-of-distribution test set data. We measure the detection performance using ResNet trained on CIFAR-10, when SVHN dataset is used as OOD. All values are percentages and the best results are indicated in bold.

## 3 Experimental results

In this section, we demonstrate the effectiveness of the proposed method using deep convolutional neural networks such as DenseNet [14] and ResNet [12] on various vision datasets: CIFAR [15], SVHN [28], ImageNet [5] and LSUN [32]. Due to the space limitation, we provide the more detailed experimental setups and results in the supplementary material. Our code is available at https://github.com/pokaxpoka/deep_Mahalanobis_detector.

### 3.1 Detecting out-of-distribution samples

**Setup.** For the problem of detecting out-of-distribution (OOD) samples, we train DenseNet with 100 layers and ResNet with 34 layers for classifying CIFAR-10, CIFAR-100 and SVHN datasets. The dataset used in training is the in-distribution (positive) dataset and the others are considered as OOD (negative). We only use test datasets for evaluation. In addition, the TinyImageNet (i.e., subset of ImageNet dataset) and LSUN datasets are also tested as OOD. For evaluation, we use a threshold-based detector which measures some confidence score of the test sample, and then classifies the test sample as in-distribution if the confidence score is above some threshold. We measure the following metrics: the true negative rate (TNR) at 95% true positive rate (TPR), the area under the receiver operating characteristic curve (AUROC), the area under the precision-recall curve (AUPR), and the detection accuracy. For comparison, we consider the baseline method [13], which defines a confidence score as a maximum value of the posterior distribution, and the state-of-the-art ODIN [21], which defines the confidence score as a maximum value of the processed posterior distribution.

For our method, we extract the confidence scores from every end of dense (or residual) block of DenseNet (or ResNet). The size of feature maps on each convolutional layers is reduced by average pooling for computational efficiency: $\mathcal{F} \times \mathcal{H} \times \mathcal{W} \rightarrow \mathcal{F} \times 1$, where $\mathcal{F}$ is the number of channels and $\mathcal{H} \times \mathcal{W}$ is the spatial dimension. As shown in Algorithm 1, the output of the logistic regression detector is used as the final confidence score in our case. All hyperparameters are tuned on a separate validation set, which consists of 1,000 images from each in- and out-of-distribution pair. Similar to Ma et al. [22], the weights of logistic regression detector are trained using nested cross validation within the validation set, where the class label is assigned positive for in-distribution samples and assigned negative for OOD samples. Since one might not have OOD validation datasets in practice, we also consider tuning the hyperparameters using in-distribution (positive) samples and corresponding adversarial (negative) samples generated by FGSM [10].

**Contribution by each technique and comparison with ODIN.** Table 1 validates the contributions of our suggested techniques under the comparison with the baseline method and ODIN. We measure the detection performance using ResNet trained on CIFAR-10, when SVHN dataset is used as OOD. We incrementally apply our techniques to see the stepwise improvement by each component. One can note that our method significantly outperforms the baseline method without feature ensembles and input pre-processing. This implies that our method can characterize the OOD samples very effectively compared to the posterior distribution. By utilizing the feature ensemble and input pre-processing, the detection performance are further improved compared to that of ODIN. The left-hand column of Table 2 reports the detection performance with ODIN for all in- and out-of-distribution

| In-dist (model) | OOD | Validation on OOD samples | | | Validation on adversarial samples | | |
|---|---|---|---|---|---|---|---|
| | | TNR at TPR 95% | AUROC | Detection acc. | TNR at TPR 95% | AUROC | Detection acc. |
| | | Baseline [13] / ODIN [21] / Mahalanobis (ours) | | | Baseline [13] / ODIN [21] / Mahalanobis (ours) | | |
| CIFAR-10 (DenseNet) | SVHN | 40.2 / 86.2 / **90.8** | 89.9 / 95.5 / **98.1** | 83.2 / 91.4 / **93.9** | 40.2 / 70.5 / **89.6** | 89.9 / 92.8 / **97.6** | 83.2 / 86.5 / **92.6** |
| | TinyImageNet | 58.9 / 92.4 / **95.0** | 94.1 / 98.5 / **98.8** | 88.5 / 93.9 / **95.0** | 58.9 / 87.1 / **94.9** | 94.1 / 97.2 / **98.8** | 88.5 / 92.1 / **95.0** |
| | LSUN | 66.6 / 96.2 / **97.2** | 95.4 / 99.2 / **99.3** | 90.3 / 95.7 / **96.3** | 66.6 / 92.9 / **97.2** | 95.4 / 98.5 / **99.2** | 90.3 / 94.3 / **96.2** |
| CIFAR-100 (DenseNet) | SVHN | 26.7 / 70.6 / **82.5** | 82.7 / 93.8 / **97.2** | 75.6 / 86.6 / **91.5** | 26.7 / 39.8 / **62.2** | 82.7 / 88.2 / **91.8** | 75.6 / 80.7 / **84.6** |
| | TinyImageNet | 17.6 / 42.6 / **86.6** | 71.7 / 85.2 / **97.4** | 65.7 / 77.0 / **92.2** | 17.6 / 43.2 / **87.2** | 71.7 / 85.3 / **97.0** | 65.7 / 77.2 / **91.8** |
| | LSUN | 16.7 / 41.2 / **91.4** | 70.8 / 85.5 / **98.0** | 64.9 / 77.1 / **93.9** | 16.7 / 42.1 / **91.4** | 70.8 / 85.7 / **97.9** | 64.9 / 77.3 / **93.8** |
| SVHN (DenseNet) | CIFAR-10 | 69.3 / 71.7 / **96.8** | 91.9 / 91.4 / **98.9** | 86.6 / 85.8 / **95.9** | 69.3 / 69.3 / **97.5** | 91.9 / 91.9 / **98.8** | 86.6 / 86.6 / **96.3** |
| | TinyImageNet | 79.8 / 84.1 / **99.9** | 94.8 / 95.1 / **99.9** | 90.2 / 90.4 / **98.9** | 79.8 / 79.8 / **99.9** | 94.8 / 94.8 / **99.8** | 90.2 / 90.2 / **98.9** |
| | LSUN | 77.1 / 81.1 / **100** | 94.1 / 94.5 / **99.9** | 89.1 / 89.2 / **99.3** | 77.1 / 77.1 / **100** | 94.1 / 94.1 / **99.9** | 89.1 / 89.1 / **99.2** |
| CIFAR-10 (ResNet) | SVHN | 32.5 / 86.6 / **96.4** | 89.9 / 96.7 / **99.1** | 85.1 / 91.1 / **95.8** | 32.5 / 40.3 / **75.8** | 89.9 / 86.5 / **95.5** | 85.1 / 77.8 / **89.1** |
| | TinyImageNet | 44.7 / 72.5 / **97.1** | 91.0 / 94.0 / **99.5** | 85.1 / 86.5 / **96.3** | 44.7 / 69.6 / **95.5** | 91.0 / 93.9 / **99.0** | 85.1 / 86.0 / **95.4** |
| | LSUN | 45.4 / 73.8 / **98.9** | 91.0 / 94.1 / **99.7** | 85.3 / 86.7 / **97.7** | 45.4 / 70.0 / **98.1** | 91.0 / 93.7 / **99.5** | 85.3 / 85.8 / **97.2** |
| CIFAR-100 (ResNet) | SVHN | 20.3 / 62.7 / **91.9** | 79.5 / 93.9 / **98.4** | 73.2 / 88.0 / **93.7** | 20.3 / 12.2 / **41.9** | 79.5 / 72.0 / **84.4** | 73.2 / 67.7 / **76.5** |
| | TinyImageNet | 20.4 / 49.2 / **90.9** | 77.2 / 87.6 / **98.2** | 70.8 / 80.1 / **93.3** | 20.4 / 33.5 / **70.3** | 77.2 / 83.6 / **87.9** | 70.8 / 75.9 / **84.6** |
| | LSUN | 18.8 / 45.6 / **90.9** | 75.8 / 85.6 / **98.2** | 69.9 / 78.3 / **93.5** | 18.8 / 31.6 / **56.6** | 75.8 / 81.9 / **82.3** | 69.9 / 74.6 / **79.7** |
| SVHN (ResNet) | CIFAR-10 | 78.3 / 79.8 / **98.4** | 92.9 / 92.1 / **99.3** | 90.0 / 89.4 / **96.9** | 78.3 / 79.8 / **94.1** | 92.9 / 92.1 / **97.6** | 90.0 / 89.4 / **94.6** |
| | TinyImageNet | 79.0 / 82.1 / **99.9** | 93.5 / 92.0 / **99.9** | 90.4 / 89.4 / **99.1** | 79.0 / 80.5 / **99.2** | 93.5 / 92.9 / **99.3** | 90.4 / 90.1 / **98.8** |
| | LSUN | 74.3 / 77.3 / **99.9** | 91.6 / 89.4 / **99.9** | 89.0 / 87.2 / **99.5** | 74.3 / 76.3 / **99.9** | 91.6 / 90.7 / **99.9** | 89.0 / 88.2 / **99.5** |

Table 2: Distinguishing in- and out-of-distribution test set data for image classification under various validation setups. All values are percentages and the best results are indicated in bold.

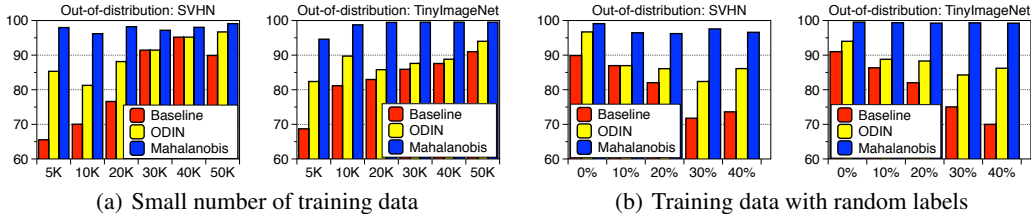

(a) Small number of training data  (b) Training data with random labels

Figure 3: Comparison of AUROC (%) under extreme scenarios: (a) small number of training data, where the x-axis represents the number of training data. (b) Random label is assigned to training data, where the x-axis represents the percentage of training data with random label.

dataset pairs. Our method outperforms the baseline and ODIN for all tested cases. In particular, our method improves the TNR, i.e., the fraction of detected LSUN samples, compared to ODIN: $41.2\% \rightarrow 91.4\%$ using DenseNet, when 95% of CIFAR-100 samples are correctly detected.

**Comparison of robustness.** In order to evaluate the robustness of our method, we measure the detection performance when all hyperparameters are tuned only using in-distribution and adversarial samples generated by FGSM [10]. As shown in the right-hand column of Table 2, ODIN is working poorly compared to the baseline method in some cases (e.g., DenseNet trained on SVHN), while our method still outperforms the baseline and ODIN consistently. We remark that our method validated without OOD but adversarial samples even outperforms ODIN validated with OOD. We also verify the robustness of our method under various training setups. Since our method utilizes empirical class mean and covariance of training samples, there is a caveat such that it can be affected by the properties of training data. In order to verify the robustness, we measure the detection performance when we train ResNet by varying the number of training data and assigning random label to training data on CIFAR-10 dataset. As shown in Figure 3, our method (blue bar) maintains high detection performances even for small number of training data or noisy one, while baseline (red bar) and ODIN (yellow bar) do not. Finally, we remark that our method using softmax neural classifier trained by standard cross entropy loss typically outperforms the ODIN using softmax neural classifier trained by confidence loss [20] which involves jointly training a generator and a classifier to calibrate the posterior distribution even though training such model is computationally more expensive (see the supplementary material for more details).

## 3.2 Detecting adversarial samples

**Setup.** For the problem of detecting adversarial samples, we train DenseNet and ResNet for classifying CIFAR-10, CIFAR-100 and SVHN datasets, and the corresponding test dataset is used as the

| Model | Dataset (model) | Score | Detection of known attack | | | | Detection of unknown attack | | | |
|---|---|---|---|---|---|---|---|---|---|---|
| | | | FGSM | BIM | DeepFool | CW | FGSM (seen) | BIM | DeepFool | CW |
| DenseNet | CIFAR-10 | KD+PU [7] | 85.96 | 96.80 | 68.05 | 58.72 | 85.96 | 3.10 | 68.34 | 53.21 |
| | | LID [22] | 98.20 | 99.74 | **85.14** | 80.05 | 98.20 | 94.55 | 70.86 | 71.50 |
| | | Mahalanobis (ours) | **99.94** | **99.78** | 83.41 | **87.31** | **99.94** | **99.51** | **83.42** | **87.95** |
| | CIFAR-100 | KD+PU [7] | 90.13 | 89.69 | 68.29 | 57.51 | 90.13 | 66.86 | 65.30 | 58.08 |
| | | LID [22] | 99.35 | 98.17 | 70.17 | 73.37 | 99.35 | 68.62 | 69.68 | 72.36 |
| | | Mahalanobis (ours) | **99.86** | **99.17** | **77.57** | **87.05** | **99.86** | **98.27** | **75.63** | **86.20** |
| | SVHN | KD+PU [7] | 86.95 | 82.06 | 89.51 | 85.68 | 86.95 | 83.28 | 84.38 | 82.94 |
| | | LID [22] | 99.35 | 94.87 | 91.79 | 94.70 | 99.35 | 92.21 | 80.14 | 85.09 |
| | | Mahalanobis (ours) | **99.85** | **99.28** | **95.10** | **97.03** | **99.85** | **99.12** | **93.47** | **96.95** |
| ResNet | CIFAR-10 | KD+PU [7] | 81.21 | 82.28 | 81.07 | 55.93 | 83.51 | 16.16 | 76.80 | 56.30 |
| | | LID [22] | 99.69 | 96.28 | 88.51 | 82.23 | 99.69 | 95.38 | 71.86 | 77.53 |
| | | Mahalanobis (ours) | **99.94** | **99.57** | **91.57** | **95.84** | **99.94** | **98.91** | **78.06** | **93.90** |
| | CIFAR-100 | KD+PU [7] | 89.90 | 83.67 | 80.22 | 77.37 | 89.90 | 68.85 | 57.78 | 73.72 |
| | | LID [22] | 98.73 | 96.89 | 71.95 | 78.67 | 98.73 | 55.82 | 63.15 | 75.03 |
| | | Mahalanobis (ours) | **99.77** | **96.90** | **85.26** | **91.77** | **99.77** | **96.38** | **81.95** | **90.96** |
| | SVHN | KD+PU [7] | 82.67 | 66.19 | 89.71 | 76.57 | 82.67 | 43.21 | **84.30** | 67.85 |
| | | LID [22] | 97.86 | 90.74 | 92.40 | 88.24 | 97.86 | 84.88 | 67.28 | 76.58 |
| | | Mahalanobis (ours) | **99.62** | **97.15** | **95.73** | **92.15** | **99.62** | **95.39** | 72.20 | **86.73** |

Table 3: Comparison of AUROC (%) under various validation setups. For evaluation on unknown attack, FGSM samples denoted by "seen" are used for validation. For our method, we use both feature ensemble and input pre-processing. The best results are indicated in bold.

positive samples to measure the performance. We use adversarial images as the negative samples generated by the following attack methods: FGSM [10], BIM [16], DeepFool [26] and CW [3], where the detailed explanations can be found in the supplementary material. For comparison, we use a logistic regression detector based on combinations of kernel density (KD) [7] and predictive uncertainty (PU), i.e., maximum value of posterior distribution. We also compare the state-of-the-art local intrinsic dimensionality (LID) scores [22]. Following the similar strategies in [7, 22], we randomly choose 10% of original test samples for training the logistic regression detectors and the remaining test samples are used for evaluation. Using nested cross-validation within the training set, all hyper-parameters are tuned.

**Comparison with LID and generalization analysis.** The left-hand column of Table 3 reports the AUROC score of a logistic regression detectors for all normal and adversarial pairs. One can note that the proposed method outperforms all tested methods in most cases. In particular, ours improves the AUROC of LID from $82.2\%$ to $95.8\%$ when we detect CW samples using ResNet trained on the CIFAR-10 dataset. Similar to [22], we also evaluate whether the proposed method is tuned on a simple attack can be generalized to detect other more complex attacks. To this end, we measure the detection performance when we train the logistic regression detector using samples generated by FGSM. As shown in the right-hand column of Table 3, our method trained on FGSM can accurately detect much more complex attacks such as BIM, DeepFool and CW. Even though LID can also generalize well, our method still outperforms it in most cases. A natural question that arises is whether the LID can be useful in detecting OOD samples. We indeed compare the performance of our method with that of LID in the supplementary material, where our method still outperforms LID in all tested case.

### 3.3 Class-incremental learning

**Setup.** For the task of class-incremental learning, we train ResNet with 34 layers for classifying CIFAR-100 and downsampled ImageNet [4]. As described in Section 2.3, we assume that a classifier is pre-trained on a certain amount of base classes and new classes with corresponding datasets are incrementally provided one by one. Specifically, we test two different scenarios: in the first scenario, half of CIFAR-100 classes are bases classes and the rest are new classes. In the second scenario, all classes in CIFAR-100 are considered to be base classes and 100 of ImageNet classes are new classes. All scenarios are tested five times and then averaged. Class splits are randomly generated for each trial. For comparison, we consider a softmax classifier, which is fine-tuned whenever new class data come in, and a Euclidean classifier [25], which tries to accommodate a new class by only computing the class mean. For the softmax classifier, we only update the softmax layer to achieve near-zero cost training [25], and follow the memory management in Rebuffi & Kolesnikov [29]: a small number of samples from old classes are kept in the limited memory, where the size of the

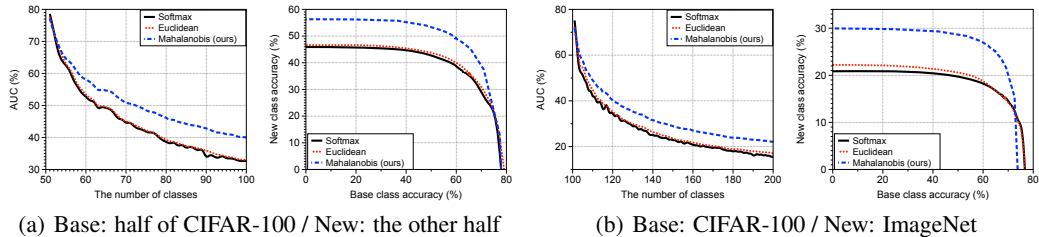

(a) Base: half of CIFAR-100 / New: the other half      (b) Base: CIFAR-100 / New: ImageNet

Figure 4: Experimental results of class-incremental learning on CIFAR-100 and ImageNet datasets. In each experiment, we report (left) AUC with respect to the number of learned classes and, (right) the base-new class accuracy curve after the last new classes is added.

memory is matched with that for keeping the parameters for Mahalanobis distance-based classifier. Namely, the number of old exemplars kept for training the softmax classifier is chosen as the sum of the number of learned classes and the dimension (512 in our experiments) of the hidden features. For evaluation, similar to [18], we first draw base-new class accuracy curve by adjusting an additional bias to the new class scores, and measure the area under curve (AUC) since averaging base and new class accuracy may cause an imbalanced measure of the performance between base and new classes.

**Comparison with other classifiers.** Figure 4 compares the incremental learning performance of methods in terms of AUC in the two scenarios mentioned above. In each sub-figure, AUC with respect to the number of learned classes (left) and the base-new class accuracy curve after the last new classes is added (right) are drawn. Our proposed Mahalanobis distance-based classifier outperforms the other methods by a significant margin, as the number of new classes increases, although there is a crossing in the right figure of Figure 4(b) in small regimes (due to the catastrophic forgetting issue). In particular, the AUC of our proposed method is 40.0% (22.1%), which is better than 32.7% (15.6%) of the softmax classifier and 32.9% (17.1%) of the Euclidean distance classifier after all new classes are added in the first (second) experiment. We also report the experimental results in the supplementary material for the case when classes of CIFAR-100 are base classes and those of CIFAR-10 are new classes, where the overall trend is similar. The experimental results additionally demonstrate the superiority of our confidence score, compared to other plausible ones.

# 4   Conclusion

In this paper, we propose a simple yet effective method for detecting abnormal test samples including both out-of-distribution and adversarial ones. In essence, our main idea is inducing a generative classifier under LDA assumption, and defining new confidence score based on it. With calibration techniques such as input pre-processing and feature ensemble, our method performs very strongly across multiple tasks: detecting out-of-distribution samples, detecting adversarial attacks and class-incremental learning. We also found that our proposed method is more robust in the choice of its hyperparameters as well as against extreme scenarios, e.g., when the training dataset has some noisy, random labels or a small number of data samples. We believe that our approach have a potential to apply to many other related machine learning tasks, e.g., active learning [8], ensemble learning [19] and few-shot learning [31].

**Acknowledgements**

This work was supported in part by Institute for Information & communications Technology Promotion (IITP) grant funded by the Korea government (MSIT) (No.R0132-15-1005, Content visual browsing technology in the online and offline environments), National Research Council of Science & Technology (NST) grant by the Korea government (MSIP) (No. CRC-15-05-ETRI), DARPA Explainable AI (XAI) program #313498, Sloan Research Fellowship, and Kwanjeong Educational Foundation Scholarship.

## Footnotes

[1]For example, state-of-the-art CNNs trained on large-scale image dataset are off-the-shelf [12, 14], so they are a starting point in many computer vision tasks [9, 18, 25].

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
