[Supplementary Material]

# Supplementary Material:

# A Simple Unified Framework for Detecting Out-of-Distribution Samples and Adversarial Attacks

## A    Preliminaries for Gaussian discriminant analysis

In this section, we describe the basic concept of the discriminative and generative classifier [27]. Formally, denote the random variable of the input and label as $\mathbf{x} \in \mathcal{X}$ and $y \in \mathcal{Y} = \{1, \cdots, C\}$, respectively. For the classification task, the discriminative classifier directly defines a posterior distribution $P(y|\mathbf{x})$, i.e., learning a direct mapping between input $\mathbf{x}$ and label $y$. A popular model for discriminative classifier is softmax classifier which defines the posterior distribution as follows: $P\left(y = c|\mathbf{x}\right) = \frac{\exp\left(\mathbf{w}_c^\top \mathbf{x} + b_c\right)}{\sum_{c'} \exp\left(\mathbf{w}_{c'}^\top \mathbf{x} + b_{c'}\right)}$, where $\mathbf{w}_c$ and $b_c$ are weights and bias for a class $c$, respectively. In contrast to the discriminative classifier, the generative classifier defines the class conditional distribution $P\left(\mathbf{x}|y\right)$ and class prior $P\left(y\right)$ in order to indirectly define the posterior distribution by specifying the joint distribution $P\left(\mathbf{x}, y\right) = P\left(y\right) P\left(\mathbf{x}|y\right)$. Gaussian discriminant analysis (GDA) is a popular method to define the generative classifier by assuming that the class conditional distribution follows the multivariate Gaussian distribution and the class prior follows Bernoulli distribution: $P\left(\mathbf{x}|y = c\right) = \mathcal{N}\left(\mathbf{x}|\mu_c, \mathbf{\Sigma}_c\right), P\left(y = c\right) = \frac{\beta_c}{\sum_{c'} \beta_{c'}}$, where $\mu_c$ and $\mathbf{\Sigma}_c$ are the mean and covariance of multivariate Gaussian distribution, and $\beta_c$ is the unnormalized prior for class $c$. This classifier has been studied in various machine learning areas (e.g., semi-supervised learning [17] and incremental learning [29]).

In this paper, we focus on the special case of GDA, also known as the linear discriminant analysis (LDA). In addition to Gaussian assumption, LDA further assumes that all classes share the same covariance matrix, i.e., $\mathbf{\Sigma}_c = \mathbf{\Sigma}$. Since the quadratic term is canceled out with this assumption, the posterior distribution of generative classifier can be represented as follows:

$$P\left(y = c|\mathbf{x}\right) = \frac{P\left(y = c\right) P\left(\mathbf{x}|y = c\right)}{\sum_{c'} P\left(y = c'\right) P\left(\mathbf{x}|y = c'\right)} = \frac{\exp\left(\mu_c^\top \mathbf{\Sigma}^{-1} \mathbf{x} - \frac{1}{2} \mu_c^\top \mathbf{\Sigma}^{-1} \mu_c + \log \beta_c\right)}{\sum_{c'} \exp\left(\mu_{c'}^\top \mathbf{\Sigma}^{-1} \mathbf{x} - \frac{1}{2} \mu_{c'}^\top \mathbf{\Sigma}^{-1} \mu_{c'} + \log \beta_{c'}\right)}.$$

One can note that the above form of posterior distribution is equivalent to the softmax classifier by considering $\mu_c^\top \mathbf{\Sigma}^{-1}$ and $-\frac{1}{2} \mu_c^\top \mathbf{\Sigma}^{-1} \mu_c + \log \beta_c$ as weight and bias of it, respectively. This implies that $\mathbf{x}$ might be fitted in Gaussian distribution during training a softmax classifier.

## B    Experimental setup

In this section, we describe detailed explanation about all the experiments described in Section 3.

### B.1    Experimental setups in detecting out-of-distribution

**Detailed model architecture and training.** We consider two state-of-the-art neural network architectures: DenseNet [14] and ResNet [12]. For DenseNet, our model follows the same setup as in Huang & Liu [14]: 100 layers, growth rate $k = 12$ and dropout rate 0. Also, we use ResNet with 34 layers and dropout rate 0.[2] The softmax classifier is used, and each model is trained by minimizing the cross-entropy loss using SGD with Nesterov momentum. Specifically, we train DenseNet for 300 epochs with batch size 64 and momentum 0.9. For ResNet, we train it for 200 epochs with batch size 128 and momentum 0.9. The learning rate starts at 0.1 and is dropped by a factor of 10 at 50% and 75% of the training progress, respectively. The test set errors of DenseNet and ResNet on CIFAR-10, CIFAR-100 and SVHN are reported in Table 4.

**Datasets.** We train DenseNet and ResNet for classifying CIFAR-10 (or 100) and SVHN datasets: the former consists of 50,000 training and 10,000 test images with 10 (or 100) image classes, and the latter consists of 73,257 training and 26,032 test images with 10 digits.[3] The corresponding

test dataset is used as the in-distribution (positive) samples to measure the performance. We use realistic images as the out-of-distribution (negative) samples: the TinyImageNet consists of 10,000 test images with 200 image classes from a subset of ImageNet images. The LSUN consists of 10,000 test images of 10 different scenes. We downsample each image of TinyImageNet and LSUN to size $32 \times 32$.[4]

**Tested methods.** In this paper, we consider the baseline method [13] and ODIN [21] for comparison. The confidence score in Hendrycks & Gimpel [13] is a maximum value of softmax posterior distribution, i.e., $\max_y P(y|\mathbf{x})$. The key idea of ODIN is the temperature scaling which is defined as follows:

$$P(y = \widehat{y}|\mathbf{x}; T) = \frac{\exp\left(f_{\widehat{y}}(\mathbf{x})/T\right)}{\sum_y \exp\left(f_y(\mathbf{x})/T\right)},$$

where $T > 0$ is the temperature scaling parameter and $\mathbf{f} = (f_1, \ldots, f_K)$ is final feature vector of deep neural networks. For each data $\mathbf{x}$, ODIN first calculates the pre-processed image $\widehat{\mathbf{x}}$ by adding the small perturbations as follows:

$$\mathbf{x}' = \mathbf{x} - \varepsilon_{odin}\text{sign}\left(- \bigtriangledown_{\mathbf{x}} \log P_\theta(y = \widehat{y}|\mathbf{x}; T)\right),$$

where $\varepsilon_{odin}$ is a magnitude of noise and $\widehat{y}$ is the predicted label. Next, ODIN feeds the pre-processed data into the classifier, computes the maximum value of scaled predictive distribution, i.e., $\max_y P_\theta(y|\mathbf{x}'; T)$, and classifies it as positive (i.e., in-distribution) if the confidence score is above some threshold $\delta$. For ODIN, the perturbation noise $\varepsilon_{odin}$ is chosen from $\{0, 0.0005, 0.001, 0.0014, 0.002, 0.0024, 0.005, 0.01, 0.05, 0.1, 0.2\}$, and the temperature $T$ is chosen from $\{1, 10, 100, 1000\}$.

**Hyper parameters for our method.** There are two hyper parameters in our method: the magnitude of noise in (4) and layer indexes for feature ensemble. For all experiments, we extract the confidence scores from every end of dense (or residual) block of DenseNet (or ResNet). The size of feature maps on each convolutional layers is reduced by average pooling for computational efficiency: $\mathcal{F} \times \mathcal{H} \times \mathcal{W} \rightarrow \mathcal{F} \times 1$, where $\mathcal{F}$ is the number of channels and $\mathcal{H} \times \mathcal{W}$ is the spatial dimension. The magnitude of noise in (4) is chosen from $\{0, 0.0005, 0.001, 0.0014, 0.002, 0.0024, 0.005, 0.01, 0.05, 0.1, 0.2\}$.

**Performance metrics.** For evaluation, we measure the following metrics to measure the effectiveness of the confidence scores in distinguishing in- and out-of-distribution images.

- **True negative rate (TNR) at 95% true positive rate (TPR).** Let TP, TN, FP, and FN denote true positive, true negative, false positive and false negative, respectively. We measure TNR = TN / (FP+TN), when TPR = TP / (TP+FN) is 95%.

- **Area under the receiver operating characteristic curve (AUROC).** The ROC curve is a graph plotting TPR against the false positive rate = FP / (FP+TN) by varying a threshold.

- **Area under the precision-recall curve (AUPR).** The PR curve is a graph plotting the precision = TP / (TP+FP) against recall = TP / (TP+FN) by varying a threshold. AUPR-IN (or -OUT) is AUPR where in- (or out-of-) distribution samples are specified as positive.

- **Detection accuracy.** This metric corresponds to the maximum classification probability over all possible thresholds $\delta$:

$$1 - \min_\delta \left\{ P_{\texttt{in}} \left(q\left(\mathbf{x}\right) \le \delta\right) P\left(\mathbf{x} \text{ is from } P_{\texttt{in}}\right) + P_{\texttt{out}} \left(q\left(\mathbf{x}\right) > \delta\right) P\left(\mathbf{x} \text{ is from } P_{\texttt{out}}\right) \right\},$$

where $q(\mathbf{x})$ is a confident score. We assume that both positive and negative examples have equal probability of appearing in the test set, i.e., $P\left(\mathbf{x} \text{ is from } P_{\texttt{in}}\right) = P\left(\mathbf{x} \text{ is from } P_{\texttt{out}}\right)$.

Note that AUROC, AUPR and detection accuracy are threshold-independent evaluation metrics.

### B.2 Experimental setups in detecting adversarial samples

**Adversarial attacks.** For the problem of detecting adversarial samples, we consider the following attack methods: fast gradient sign method (FGSM) [10], basic iterative method (BIM) [16], Deep-Fool [26] and Carlini-Wagner (CW) [3]. The FGSM directly perturbs normal input in the direction of the loss gradient. Formally, non-targeted adversarial examples are constructed as

$$\mathbf{x}_{adv} = \mathbf{x} + \varepsilon_{FGSM}\text{sign}\left(\bigtriangledown_{\mathbf{x}}\ell(y^*, P(y|\mathbf{x}))\right),$$

|  |  | CIFAR-10 | | CIFAR-100 | | SVHN | |
|---|---|---|---|---|---|---|---|
|  |  | $L_\infty$ | Acc. | $L_\infty$ | Acc. | $L_\infty$ | Acc. |
| DenseNet | Clean | 0 | 95.19% | 0 | 77.63% | 0 | 96.38% |
|  | FGSM | 0.21 | 20.04% | 0.21 | 4.86% | 0.21 | 56.27% |
|  | BIM | 0.22 | 0.00% | 0.22 | 0.02% | 0.22 | 0.67% |
|  | DeepFool | 0.30 | 0.23% | 0.25 | 0.23% | 0.57 | 0.50% |
|  | CW | 0.05 | 0.10% | 0.03 | 0.16% | 0.12 | 0.54% |
| ResNet | Clean | 0 | 93.67% | 0 | 78.34% | 0 | 96.68% |
|  | FGSM | 0.25 | 23.98% | 0.25 | 11.67% | 0.25 | 49.33% |
|  | BIM | 0.26 | 0.02% | 0.26 | 0.21% | 0.26 | 2.37% |
|  | DeepFool | 0.36 | 0.33% | 0.27 | 0.37% | 0.62 | 13.20% |
|  | CW | 0.08 | 0.00% | 0.08 | 0.01% | 0.15 | 0.04% |

Table 4: The $L_\infty$ mean perturbation and classification accuracy on clean and adversarial samples.

where $\varepsilon_{FGSM}$ is a magnitude of noise, $y^*$ is the ground truth label and $\ell$ is a loss function to measure the distance between the prediction and the ground truth. The BIM is an iterative version of FGSM, which applies FGSM multiple times with a smaller step size. Formally, non-targeted adversarial examples are constructed as

$$\mathbf{x}_{adv}^0 = \mathbf{x}, \ \mathbf{x}_{adv}^{n+1} = \text{Clip}_{\mathbf{x}}^{\varepsilon_{BIM}} \{\mathbf{x}_{adv}^n + \alpha_{BIM}\text{sign}\left(\nabla_{\mathbf{x}_{adv}^n}\ell(y^*, P(y|\mathbf{x}_{adv}^n))\right)\},$$

where $\text{Clip}_{\mathbf{x}}^{\varepsilon_{BIM}}$ means we clip the resulting image to be within the $\varepsilon_{BIM}$-ball of $\mathbf{x}$. DeepFool works by finding the closest adversarial examples with geometric formulas. CW is an optimization-based method which arguably the most effective method. Formally, non-targeted adversarial examples are constructed as

$$\arg\min_{\mathbf{x}_{adv}} \ \lambda d(\mathbf{x}, \mathbf{x}_{adv}) - \ell(y^*, P(y|\mathbf{x}_{adv})),$$

where $\lambda$ is penalty parameter and $d(\cdot, \cdot)$ is a metric to quantify the distance between an original image and its adversarial counterpart. However, compared to FGSM and BIM, this method is much slower in practice. For all experiments, $L_2$ distance is used as a constraint. We used the library from FaceBook [11] for generating adversarial samples.[5] Table 4 tatistics of adversarial attacks including the $L_\infty$ mean perturbation and classification accuracy on adversarial attacks.

**Tested methods.** Ma et al. [22] proposed to characterize adversarial subspaces by using local intrinsic dimensionality (LID). Given a test sample $\mathbf{x}$, LID is defined as follows:

$$\widehat{LID} = -\left(\frac{1}{k}\sum_i \log \frac{r_i(\mathbf{x})}{r_k(\mathbf{x})}\right),$$

where $r_i(\mathbf{x})$ denotes the distance between $\mathbf{x}$ and its $i$-th nearest neighbor within a sample of points drawn from in-distribution, and $r_k(\mathbf{x})$ denotes the maximum distance among $k$ nearest neighbors. We commonly extract the LID scores from every end of dense (or residual) block of DenseNet (or ResNet) similar to ours. Given test sample $\mathbf{x}$ and the set $\mathbf{X}_c$ of training samples with label $c$, the Gaussian kernel density with bandwidth $\sigma$ is defined as follows:

$$KD(\mathbf{x}) = \frac{1}{|\mathbf{X}_c|}\sum_{\mathbf{x}_i \in \mathbf{X}_c} k_\sigma(\mathbf{x}_i, \mathbf{x}),$$

where $k_\sigma(x, y) \propto \exp(-||x - y||^2/\sigma^2)$. For LID and KD, we used the library from Ma et al. [22].

**Hyper-parameters and training.** Following the similar strategies in [7, 22], we randomly choose 10% of original test samples for training the logistic regression detectors and the remaining test samples are used for evaluation. The training sets consists of three types of examples: adversarial, normal and noisy. Here, noisy examples are generated by adding random noise to normal examples. Using nested cross validation within the training set, all hyper-parameters including the bandwidth

(a) Small training data: the x-axis represents the number of training data

(b) Noisy training data: the x-axis represents the percentage of training data with random label

Figure 5: Comparison of AUROC (%) under different training data. To evaluate the robustness of proposed method, we train ResNet (a) by varying the number of training data and (b) assigning random label to training data on CIFAR-10 dataset.

(a) Base: CIFAR-100 / New: CIFAR-10

Figure 6: Experimental results of class-incremental learning on CIFAR-100 and CIFAR-10 datasets. We report (left) AUC with respect to the number of learned classes and, (right) the base-new class accuracy curve after the last new classes is added.

parameter for KD, the number of nearest neighbors for LID, and input noise for our method are tuned. Specifically, the value of $k$ is chosen from $\{10, 20, 30, 40, 50, 60, 70, 80, 90\}$ with respect to a minibatch of size 100, and the bandwidth was chosen from $\{0.1, 0.25, 0.5, 0.75, 1\}$. The magnitude of noise in (4) is chosen from $\{0, 0.0005, 0.001, 0.0014, 0.002, 0.0024, 0.005, 0.01, 0.05, 0.1, 0.2\}$.

## C More experimental results

In this section, we provide more experimental results.

### C.1 Robustness of our method in detecting adversarial samples

In order to verify the robustness, we measure the detection performance when we train ResNet by varying the number of training data and assigning random label to training data on CIFAR-10 dataset. As shown in Figure 5, our method (blue bar) outperforms LID (green bar) for all experiments.

### C.2 Class-incremental learning

Figure 6 compares the AUCs of tested methods when CIFAR-100 is pre-trained and CIFAR-10 is used as new classes. Our proposed Mahalanobis distance-based classifier outperforms the other methods by a significant margin, as the number of new classes increases. The AUC of our proposed method is 47.7%, which is better than 41.0% of the softmax classifier and 43.0% of the Euclidean distance classifier after all new classes are added.

## C.3 Experimental results on joint confidence loss

In addition, we remark that the proposed detector using softmax neural classifier trained by standard cross entropy loss typically outperforms the ODIN detector using softmax neural classifier trained by confidence loss [19] which involves jointly training a generator and a classifier to calibrate the posterior distribution. Also, our detector provides further improvement if one use it with model trained by confidence loss. In other words, our proposed method can improve any pre-trained softmax neural classifier.

(a) In-distribution: CIFAR-10

(b) In-distribution: SVHN

Figure 7: Performances of the baseline detector [13], ODIN detector [21] and Mahalanobis detector under various training losses.

## C.4 Comparison with ODIN

| In-dist (model) | Out-of-dist | TNR at TPR 95% | AUROC | Detection accuracy | AUPR in | AUPR out |
|---|---|---|---|---|---|---|
| | | Baseline [13] / ODIN [21] / Mahalanobis (ours) | | | | |
| CIFAR-10 (DenseNet) | SVHN | 40.2 / 86.2 / **90.8** | 89.9 / 95.5 / **98.1** | 83.2 / 91.4 / **93.9** | 83.1 / 78.8 / **96.6** | 94.7 / 98.3 / **99.2** |
| | TinyImageNet | 58.9 / 92.4 / **95.0** | 94.1 / 98.5 / **98.8** | 88.5 / 93.9 / **95.0** | 92.3 / 98.5 / **98.8** | 92.3 / 98.5 / **98.8** |
| | LSUN | 66.6 / 96.2 / **97.2** | 95.4 / 99.2 / **99.3** | 90.3 / 95.7 / **96.3** | 96.5 / 99.3 / **99.3** | 94.1 / **99.2** / 99.1 |
| CIFAR-100 (DenseNet) | SVHN | 26.7 / 70.6 / **82.5** | 82.7 / 93.8 / **97.2** | 75.6 / 86.6 / **91.5** | 74.3 / 87.1 / **94.8** | 91.0 / 97.3 / **98.8** |
| | TinyImageNet | 17.6 / 42.6 / **86.6** | 71.7 / 85.2 / **97.4** | 65.7 / 77.0 / **92.2** | 74.2 / 85.6 / **97.6** | 69.0 / 84.5 / **97.2** |
| | LSUN | 16.7 / 41.2 / **91.4** | 70.8 / 85.5 / **98.0** | 64.9 / 77.1 / **93.9** | 74.1 / 86.4 / **98.2** | 67.9 / 84.2 / **97.5** |
| SVHN (DenseNet) | CIFAR-10 | 69.3 / 71.7 / **96.8** | 91.9 / 91.4 / **98.9** | 86.6 / 85.8 / **95.9** | 95.7 / 95.2 / **99.6** | 82.8 / 84.5 / **95.8** |
| | TinyImageNet | 79.8 / 84.1 / **99.9** | 94.8 / 95.1 / **99.9** | 90.2 / 90.4 / **98.9** | 97.2 / 97.1 / **99.9** | 88.4 / 91.4 / **99.6** |
| | LSUN | 77.1 / 81.1 / **100.0** | 94.1 / 94.5 / **99.9** | 89.1 / 89.2 / **99.3** | 97.0 / 97.0 / **99.9** | 87.4 / 90.5 / **99.7** |
| CIFAR-10 (ResNet) | SVHN | 32.5 / 86.6 / **96.4** | 89.9 / 96.7 / **99.1** | 85.1 / 91.1 / **95.8** | 85.4 / 92.5 / **98.3** | 94.0 / 98.5 / **99.6** |
| | TinyImageNet | 44.7 / 72.5 / **97.1** | 91.0 / 94.0 / **99.5** | 85.1 / 86.5 / **96.3** | 92.5 / 94.2 / **99.5** | 88.4 / 94.1 / **99.5** |
| | LSUN | 45.4 / 73.8 / **98.9** | 91.0 / 94.1 / **99.7** | 85.3 / 86.7 / **97.7** | 92.5 / 94.2 / **99.7** | 88.6 / 94.3 / **99.7** |
| CIFAR-100 (ResNet) | SVHN | 20.3 / 62.7 / **91.9** | 79.5 / 93.9 / **98.4** | 73.2 / 88.0 / **93.7** | 64.8 / 89.0 / **96.4** | 89.0 / 96.9 / **99.3** |
| | TinyImageNet | 20.4 / 49.2 / **90.9** | 77.2 / 87.6 / **98.2** | 70.8 / 80.1 / **93.3** | 79.7 / 87.1 / **98.2** | 73.3 / 87.4 / **98.2** |
| | LSUN | 18.8 / 45.6 / **90.9** | 75.8 / 85.6 / **98.2** | 69.9 / 78.3 / **93.5** | 77.6 / 84.5 / **98.4** | 72.0 / 85.7 / **97.8** |
| SVHN (ResNet) | CIFAR-10 | 78.3 / 79.8 / **98.4** | 92.9 / 92.1 / **99.3** | 90.0 / 89.4 / **96.9** | 95.1 / 94.0 / **99.7** | 85.7 / 86.8 / **97.0** |
| | TinyImageNet | 79.0 / 82.1 / **99.9** | 93.5 / 92.0 / **99.9** | 90.4 / 89.4 / **99.1** | 95.7 / 93.9 / **99.9** | 86.2 / 88.1 / **99.1** |
| | LSUN | 74.3 / 77.3 / **99.9** | 91.6 / 89.4 / **99.9** | 89.0 / 87.2 / **99.5** | 94.2 / 92.1 / **99.9** | 84.0 / 85.5 / **99.1** |

Table 5: Distinguishing in- and out-of-distribution test set data for image classification. We tune the hyper-parameters using validation set of in- and out-of-distributions. All values are percentages and the best results are indicated in bold.

| In-dist (model) | Out-of-dist | TNR at TPR 95% | AUROC | Detection accuracy | AUPR in | AUPR out |
|---|---|---|---|---|---|---|
| | | Baseline [13] / ODIN [21] / Mahalanobis (ours) | | | | |
| CIFAR-10 (DenseNet) | SVHN | 40.2 / 70.5 / **89.6** | 89.9 / 92.8 / **97.6** | 83.2 / 86.5 / **92.6** | 83.1 / 72.1 / **94.5** | 94.7 / 97.4 / **99.0** |
| | TinyImageNet | 58.9 / 87.1 / **94.9** | 94.1 / 97.2 / **98.8** | 88.5 / 92.1 / **95.0** | 95.3 / 94.7 / **98.7** | 92.3 / 97.0 / **98.8** |
| | LSUN | 66.6 / 92.9 / **97.2** | 95.4 / 98.5 / **99.2** | 90.3 / 94.3 / **96.2** | 96.5 / 97.7 / **99.3** | 94.1 / 98.2 / **99.2** |
| CIFAR-100 (DenseNet) | SVHN | 26.7 / 39.8 / **62.2** | 82.7 / 88.2 / **91.8** | 75.6 / 80.7 / **84.6** | 74.3 / 80.8 / **82.6** | 91.0 / 94.0 / **95.8** |
| | TinyImageNet | 17.6 / 43.2 / **87.2** | 71.7 / 85.3 / **97.0** | 65.7 / 77.2 / **91.8** | 74.2 / 85.8 / **96.2** | 69.0 / 84.7 / **97.1** |
| | LSUN | 16.7 / 42.1 / **91.4** | 70.8 / 85.7 / **97.9** | 64.9 / 77.3 / **93.8** | 74.1 / 86.7 / **98.1** | 67.9 / 84.6 / **97.6** |
| SVHN (DenseNet) | CIFAR-10 | 69.3 / 69.3 / **97.5** | 91.9 / 91.9 / **98.8** | 86.6 / 86.6 / **96.3** | 95.7 / 95.7 / **99.6** | 82.8 / 82.8 / **95.1** |
| | TinyImageNet | 79.8 / 79.8 / **99.9** | 94.8 / 94.8 / **99.8** | 90.2 / 90.2 / **98.9** | 97.2 / 97.2 / **99.9** | 88.4 / 88.4 / **99.5** |
| | LSUN | 77.1 / 77.1 / **100** | 94.1 / 94.1 / **99.9** | 89.1 / 89.1 / **99.2** | 97.0 / 97.0 / **99.9** | 87.4 / 87.4 / **99.6** |
| CIFAR-10 (ResNet) | SVHN | 32.5 / 40.3 / **75.8** | 89.9 / 86.5 / **95.5** | 85.1 / 77.8 / **89.1** | 85.4 / 77.8 / **91.0** | 94.0 / 93.7 / **98.0** |
| | TinyImageNet | 44.7 / 69.6 / **95.5** | 91.0 / 93.9 / **99.0** | 85.1 / 86.0 / **95.4** | 92.5 / 94.3 / **98.6** | 88.4 / 93.7 / **99.1** |
| | LSUN | 45.4 / 70.0 / **98.1** | 91.0 / 93.7 / **99.5** | 85.3 / 85.8 / **97.2** | 92.5 / 94.1 / **99.5** | 88.6 / 93.6 / **99.5** |
| CIFAR-100 (ResNet) | SVHN | 20.3 / 12.2 / **41.9** | 79.5 / 72.0 / **84.4** | 73.2 / 67.7 / **76.5** | 64.8 / 48.6 / **69.1** | 89.0 / 84.9 / **92.7** |
| | TinyImageNet | 20.4 / 33.5 / **70.3** | 77.2 / 83.6 / **87.9** | 70.8 / 75.9 / **84.6** | 79.7 / 84.5 / **76.8** | 73.3 / 81.7 / **90.7** |
| | LSUN | 18.8 / 31.6 / **56.6** | 75.8 / 81.9 / **82.3** | 69.9 / 74.6 / **79.7** | 77.6 / **82.1** / 70.3 | 72.0 / 80.3 / **85.3** |
| SVHN (ResNet) | CIFAR-10 | 78.3 / 79.8 / **94.1** | 92.9 / 92.1 / **97.6** | 90.0 / 89.4 / **94.6** | 95.1 / 94.0 / **98.1** | 85.7 / 86.8 / **94.7** |
| | TinyImageNet | 79.0 / 80.5 / **99.2** | 93.5 / 92.9 / **99.3** | 90.4 / 90.1 / **98.8** | 96.8 / 96.4 / **98.8** | 86.2 / 87.5 / **98.3** |
| | LSUN | 74.3 / 76.3 / **99.9** | 91.6 / 90.7 / **99.9** | 89.0 / 88.2 / **99.5** | 94.2 / 93.0 / **99.9** | 84.0 / 85.0 / **98.8** |

Table 6: Distinguishing in- and out-of-distribution test set data for image classification when we tune the hyper-parameters of ODIN and our method only using in-distribution and adversarial (FGSM) samples. All values are percentages and boldface values indicate relative the better results.

## C.5 LID for detecting out-of-distribution samples

Figure 8 and 9 shows the performance of the ODIN [21], LID [22] and Mahalanobis detector for each in- and out-of-distribution pair. We remark that the proposed method outperforms all tested methods.

(a) In-distribution: CIFAR-10

(b) In-distribution: CIFAR-100

(c) In-distribution: SVHN

Figure 8: Distinguishing in- and out-of-distribution test set data for image classification using ResNet.

(a) In-distribution: CIFAR-10

(b) In-distribution: CIFAR-100

(c) In-distribution: SVHN

Figure 9: Distinguishing in- and out-of-distribution test set data for image classification using DenseNet.

# D Evaluation on ImageNet dataset

In this section, we verify the performance of the proposed method using the ImageNet 2012 classification dataset [5] that consists of 1000 classes. The models are trained on the 1.28 million training images, and evaluated on the 50k validation images. For all experiments, we use the pre-trained ResNet [12] which is available at `https://github.com/pytorch/vision/blob/master/torchvision/models/resnet.py`. First, we measure the classification accuracy of generative classifier from the pre-trained model as follows:

$$\widehat{y}(\mathbf{x}) = \arg\min_c \left(f(\mathbf{x}) - \widehat{\mu}_c\right)^\top \widehat{\mathbf{\Sigma}}^{-1} \left(f(\mathbf{x}) - \widehat{\mu}_c\right) + \log \widehat{\beta}_c,$$

where $\widehat{\beta}_c = \frac{N_c}{N}$ is an empirical class prior. We remark that this corresponds to predicting a class label using the posterior distribution from generative with LDA assumption. Table 7 shows the top-1 classification accuracy on ImageNet 2012 dataset. One can note that the proposed generative classifier can perform reasonably well even though the softmax classifier outperforms it in all cases. However, we remark that the gap between them is decreasing as the training accuracy increases, i.e., the pre-trained model learned more strong representations.

| Model | Softmax (training) | Softmax (validation) | Generative (validation) |
|---|---|---|---|
| ResNet (101 layers) | 86.55 | 75.66 | 73.49 |
| ResNet (18 layers) | 69.06 | 68.69 | 63.32 |

Table 7: Top-1 accuracy (%) of ResNets on ImageNet 2012 dataset.

Next, we also evaluate the detection performance of the Mahalanobis distance-based detector on ImageNet 2012 dataset using ResNets with 18 layers. For evaluation, we consider the problem of

(a) FGSM

$$\text{minimize}_{\mathbf{x}} \ (f(\mathbf{x}) - \mu_c)^T \Sigma^{-1} (f(\mathbf{x}) - \mu_c)$$

(d) Scenario 2

Figure 10: (a)/(b) Distinguishing clean and adversarial samples using ResNet with 18 layers on ImageNet 2012 validation set. (c)/(d) Distinguishing clean and garbage samples using ResNet 18 layers trained on CIFAR-10 dataset.

Class 0   class 1.   class 2.   class 3.   class 4.   class 5.   class 6.   class 7.   class 8.   class 9.

Figure 11: The generated garbage sample and its target class.

detecting adversarial samples generated by FGSM [10] and BIM [16]. Similar to Section 3.2, we extract the confidence scores from every end of residual block of ResNet. Figure 10(a) and 10(b) show the performance of various detectors. One can note that the proposed Mahalanobis distance-based detector outperforms all tested methods including LID. These results imply that our method can be working well for the large-scale datasets.

## E  Adaptive attacks against Mahalanobis distance-based detector

In this section, we evaluate the robustness of our method by generating the garbage images which may fool the Mahalanobis distance-based detector in a white-box setting, i.e., one can access to the parameters of the classifier and that of the Mahalanobis distance-based detector. Here, we remark that accessing the parameters of the Mahalanobis distance-based detector, i.e., sample means and covariance, is not mild assumption since the information about training data is required to compute them. To attack our method, we generate a garbage images $\mathbf{x}_g$ by minimizing the Mahalanobis distance as follows:

$$\arg \min_{\mathbf{x}_g} (f(\mathbf{x}_g) - \widehat{\mu}_c)^{\top} \widehat{\boldsymbol{\Sigma}}^{-1} (f(\mathbf{x}_g) - \widehat{\mu}_c),$$

where $c$ is a target class. We test two different scenarios using ResNet with 34 layers trained on CIFAR-10 dataset. In the first scenario, we generate the garbage images only using a penultimate layer of DNNs. In the second scenario, we attack every end of residual block of ResNet. Figure 11 shows the generated samples by minimizing the Mahalanobis distance. Even though the generated sample looks like the random noise, it successfully fools the pre-trained classifier, i.e., it is classified as the target class. We measure the detection performance of the baseline [13], ODIN [21], LID [22] and the proposed Mahalanobis distance-based detector. As shown in Figure 10(c) and 10(d), our method can distinguish CIFAR-10 test and garbage images for both scenarios better than the tested methods. In particular, we remark that the input pre-processing is very useful in detecting such garbage samples. These results imply that our proposed method is robust to the attacks.

## F  Hybrid inference of generative and discriminative classifiers

In this paper, we show that the generative classifier can be very useful in characterizing the abnormal samples such as OOD and adversarial samples. Here, a caveat is that the generative classifier might degrade the classification performance. In order to handle this issue, we introduce a hybrid inference of generative and discriminative classifiers. Given a generative classifier with GDA assumptions, the posterior distribution of generative classifier via Bayes rule is given as:

$$P(y = c|\mathbf{x}) = \frac{P(y = c) P(\mathbf{x}|y = c)}{\sum_{c'} P(y = c') P(\mathbf{x}|y = c')} = \frac{\exp\left(\mu_c^{\top} \boldsymbol{\Sigma}^{-1} \mathbf{x} - \frac{1}{2}\mu_c^{\top} \boldsymbol{\Sigma}^{-1} \mu_c + \log \beta_c\right)}{\sum_{c'} \exp\left(\mu_{c'}^{\top} \boldsymbol{\Sigma}^{-1} \mathbf{x} - \frac{1}{2}\mu_{c'}^{\top} \boldsymbol{\Sigma}^{-1} \mu_{c'} + \log \beta_{c'}\right)}.$$

To match this with a standard softmax classifier's weights, the parameters of the generative classifier have to satisfy the following conditions:

$$\mu_c = \Sigma \mathbf{w}_c, \ \log \beta_c - 0.5\mu_c^\top \Sigma^{-1}\mu_c = b_c,$$

where $\mathbf{w}_c$ and $b_c$ are weights and bias for a class $c$, respectively. Using the empirical covariance $\widehat{\Sigma}$ as shown in (1), one can induce the parameters of another generative classifier which has same decision boundary with the softmax classifier as follows:

$$\tilde{\mu}_c = \widehat{\Sigma}\mathbf{w}_c, \tilde{\beta}_c = \frac{\exp(0.5\tilde{\mu}_c^\top \widehat{\Sigma}^{-1}\tilde{\mu}_c - b_c)}{\sum_{c'} \exp(0.5\tilde{\mu}_{c'}^\top \widehat{\Sigma}^{-1}\tilde{\mu}_{c'} - b_{c'})}.$$

Here, we normalize the $\tilde{\beta}_c$ to satisfy $\sum_c \tilde{\beta}_c = 1$. Then, using this generative classifier, we define new hybrid posterior distribution which combines the softmax- and sample-based generative classifiers:

$$P_h(y|\mathbf{x})$$

$$= \frac{\exp\left(\lambda\left(\widehat{\mu}_c^\top \widehat{\Sigma}^{-1}\mathbf{x} - 0.5\widehat{\mu}_c^\top \widehat{\Sigma}^{-1}\widehat{\mu}_c + \log\widehat{\beta}_c\right) + (1-\lambda)\left(\tilde{\mu}_c^\top \widehat{\Sigma}^{-1}\mathbf{x} - 0.5\tilde{\mu}_c^\top \widehat{\Sigma}^{-1}\tilde{\mu}_c + \log\tilde{\beta}_c\right)\right)}{\sum_{c'} \exp\left(\lambda\left(\widehat{\mu}_{c'}^\top \widehat{\Sigma}^{-1}\mathbf{x} - 0.5\widehat{\mu}_{c'}^\top \widehat{\Sigma}^{-1}\widehat{\mu}_{c'} + \log\widehat{\beta}_{c'}\right) + (1-\lambda)\left(\tilde{\mu}_{c'}^\top \widehat{\Sigma}^{-1}\mathbf{x} - 0.5\tilde{\mu}_{c'}^\top \widehat{\Sigma}^{-1}\tilde{\mu}_{c'} + \log\tilde{\beta}_{c'}\right)\right)},$$

where $\lambda \in [0, 1]$ is a hyper-parameter. This hybrid model can be interpreted as ensemble of softmax and generative classifiers, and one can expect that it can improve the classification performance. Table 8 compares the classification accuracy of softmax, generative and hybrid classifiers. One can note that the hybrid model improves the classification accuracy, where we determine the optimal tuning parameter between the two objectives using the validation set. We also remark that such hybrid model can be useful in detecting the abnormal samples, where we pursue these tasks in the future.

| Model | Dataset | Softmax | Generative | Hybrid |
|---|---|---|---|---|
| DenseNet | CIFAR-10 | 95.16 | 94.76 | 95.00 |
| | CIFAR-100 | 77.64 | 74.01 | 77.71 |
| | SVHN | 96.42 | 96.32 | 96.34 |
| ResNet | CIFAR-10 | 93.61 | 94.13 | 94.11 |
| | CIFAR-100 | 78.08 | 77.86 | 77.96 |
| | SVHN | 96.62 | 96.58 | 96.59 |

Table 8: Classification test set accuracy (%) of DenseNet and ResNet on CIFAR-10, CIFAR-100 and SVHN datasets.

## Footnotes

[2] ResNet architecture is available at `https://github.com/kuangliu/pytorch-cifar`.

[3] We do not use the extra SVHN dataset for training.

[4]LSUN and TinyImageNet datasets are available at https://github.com/ShiyuLiang/odin-pytorch.

[5]The code is available at https://github.com/facebookresearch/adversarial_image_defenses.