[Reviews · NeurIPS 2018]

Reviewer 1



The problem of the study is detecting abnormalities within deep neural networks, to detect out-of-distribution inputs, adversarial inputs, and new classes (for class incremental learning). To achieve this, the authors integrate class-conditional Gaussian distributions with a tied covariance (linear discriminant analysis) at various stages of a target neural network and construct distributions over the valid input (in-liers). They use the Mahalanobis distance measure of the Gaussian distribution as a confidence measure (proportional to the log-likelihood). They further enhance the confidence measure by taking Fast Gradient-Sign Method-style steps in the input space to increase the score. Finally, they combine the scores gathered at different layers of the neural network through a linear combination. They evaluate the performance of their method for out-of-distribution detection on CIFAR10, CIFAR100, and SVHN against SVHN, TinyImageNet, and LSUN using DenseNet and ResNet where they establish a new baseline for the state-of-the-art under a variety of difficult conditions. To evaluate the detection of adversarial examples, the authors compare against two other methods ([23] and [7] in the paper) in a supervised (samples of the attack type) and an unsupervised way (only samples of FGSM attack) and they demonstrate superior performance in this area as well. In the incremental learning part, they also demonstrate superior results over the previous work. Quality. The paper is technically sound and well-organized. The claims are supported by an exhaustive series of evaluations (half of which is in the supplemental material). Clarity. The paper is easy to follow. The previous work is somewhat limited, only reflecting on the most recent studies of this problem. Someone who's not already familiar with the relevant literature would have difficulty appreciating the work. Although, the paper is already dense in content and analysis, and including more relevant work would clearly push it beyond the length limit. The authors also omit some crucial details. For instance, they mention downsampling of the intermediate tensors for computational efficiency but to the best of my memory, nowhere do they mentioned how much downsampling is applied (or what is the dimensionality of the processed intermediate tensors). A discussion of memory and compute requirement is also missing from the analysis, which presumably could be added to the supplemental material. They also do not mention releasing the source code, which I strongly recommend for consideration. Originality. The work is clearly a novel combination of well-known techniques. To the best of my knowledge, they are the first group to demonstrate an effective application of the presented methods for these tasks. Significance. The results are important, especially in the out-of-distribution detection domain. The results of the detection of adversarial samples are unlikely to stay true for a long time; although fooling the presented system is probably more difficult than the previous work because the adversarial samples need to satisfy several constraints at multiple stages of the network. The presented method seems to be effective at addressing the concerns. They demonstrate a new baseline for the state-of-art in several areas. It is likely that more future work would be built based on the observations and results presented in the paper. ----------- Update after rebuttal. The author response has addressed my (minor) concerns.

Reviewer 2



This paper proposes a new method for detecting out-of-distribution samples and adversarial examples in classification tasks when using deep neural networks. The core idea is simple. Replace the softmax classification layer by Gaussian Discriminative Analysis. The assumption is that given a layer of the network (e.g., the penultimate layer of the net) the conditional distribution of the features given a class, follow a Gaussian distribution. The method then consists in estimating the mean of each distribution and the covariance assuming that the covariance matrix is the same for every class (similar to LDA as noted in Appendix). Then, by using the fact that they have now a Generative model (p(x/y)), the can produce a "confidence score" for a new sample. This is score is simply the Mahalanobis distance to the closest distribution (the log of the probability for the closest class). They show different applications of this framework to detect: out-of-distribution samples (in an experimental setup similar to previous work [22]), state-of-the-art adversarial attacks, and finally they introduce a way to deal with incremental learning. The major strength of the work is its simplicity. The method is quite simple. The method is backed by empirical results. All experiments show superior performance in comparison to previous work. The major issues are: -- No analysis. It is not clear whether the conditionally Gaussian distributed assumption is reasonable or not. Moreover, an augmented confidential score computed from the output at different levels of a Deepnet is proposed. But, this will be reasonable only if the output at each layer is Gaussian. Intuition says that this is not the case unless we are very deep in the net. No analysis is given but only an empirical comparison of the results by taking features at different levels (Figure 2). More analysis is needed. -- There is no justification why some choices are made. For example, why is the covariance matrix the same for every class? -- Novelty is limited. The paper builds a lot on top of [22]. There is no additional analysis. For instance, why does the input preprocessing improve the detection of out-of-distribution samples? -- Several explanations are unclear. In particular Section 2.3 (incremental learning) is not clearly explained/analyzed. It seems that the idea is to use the features of a deep neural net computed at the last layer as a way to reduce the dimension, and then learn a Gaussian mixture where one can control the number of classes in an online fashion. Is this the case? Also, more analysis to support some conclusions (l173-l174 "if we have enough base classes and corresponding samples ...") is needed. Is is possible to decide that we are in facing a new class? How could this be done? Algorithm 2, assumes that a set of samples from a new class (labeled) are given, so the method just need to updated the Gaussian mixture. -- Details for reproducibility are lacking. For instance, l192- "The size of features maps (...) reduced by average pooling for computational efficiency" Which is the final dimension? Also, is this only a matter of computational efficiency? or it is also due the fact that one needs to learn a covariance matrix? More analysis is needed. -- Experimental setup. The whole evaluation should be done assuming that you don't know how it looks the OOD set (or the adversarial attack). I mean, if you know how it looks, then it's just a typical toy problem of two class classification. I understand this is the way previous work address this problem, but It's time that this is changed. Regarding that point, the paper *does* this analysis ("comparison of robustness") and the results are pretty interesting (performance goes a little down but results are still very good). But, the experimental section needs to better present and discuss these results. This has to be (better) highlighted in the experimental section). -- Experimental setup regarding Incremental Learning (Section 3.3) is unclear (many unclear sentences). For instance, "we keep 5 exemplars for each class …. roughly matches with the size of the covariance matrix"). In fact, the whole experiment is unclear to me. Figure 4 is not commented (there are some crossing in the base class accuracy experiment). Minor comments -- How are datasets handled. Are all images the same size (cropped/resized)? --l2 "the fundamental" → "a fundamental" -- l134-l142. Input preprocessing. "small controlled noise". What is added is not noise. The sample is perturbed in a way that the confidence score is increased. This is clearly not random, so please do not refer to this to noise. --"out proposed method" → our method or the proposed method -- metrics are not defined (TNR,TPR,AUPR,AUROC,detection accuracy) (in particular how is computed the detection accuracy) -- l213 - are further → is further -- l230-l234. This is unclear. -- l248 - regression detectors → regression detector -- l251 (and others) - "the LID" → LID or "the LID method" -- l252 - the proposed method is tuned on → the proposed method tuned on -- l254 - When we train → when training -- l260 - all tested case → all tested cases -- l261 - Algorithm 2 → Algorithm~2. -- l264 - and Euclidean classifier → and an Euclidean classifier -- l273 - ImageNet → imageNet classes -- l282-l285 - Conclusions should summarize the method and the findings. **AFTER REBUTTAL** I thank the authors for their response. The authors did a good job clarifying many of the raised points in the reviews. The strength of the paper is that the proposed method is simple and does a good job in detecting out-of-distribution samples (SOTA). Nevertheless, I still think the paper lacks more analysis, *better and clearer justifications*, ablation study, better experimental setup (more controlled experiments).

Reviewer 3



Thie paper proposes a method, for detecting abnormal examples including OOD and adversarial ones. The authors propose to calculate the mean and covariance matrix of the input representations using the representations from the penultimate layer of the model. Using the above representations the authors can calculate the Mahalanobis distance as a scoring function. In addition, the authors demonstrated their approach on the Incremental Learning task, where they add new labels as outputs for the model. Overall this study is well written and the experiments are adequate, however, there are a few concerns; Do the authors evaluate their method on other distance metrics for OOD detection such as Euclidian distance? if so, do the authors also combine the representations from all the different layers and train an LR model for that? In general, I'm wondering if the results are due to the Mahalanobis distance/due to the use of the input representation/to the due to the use of an ensemble of representations/due to the use of LR classifier. The authors mentioned they train the LR model using adversarial examples as OOD samples ("we also apply another validation strategy where adversarial samples are used as OOD samples"). From which dataset did you get the adv examples? From the in-distribution examples? Moreover, it is not clear to me how did the authors generate and calc the results for the "Validation on adversarial samples" part in Table 2. What model did they use to generate these examples? It seems like the results for the baseline are not changing and for ODIN they stay the same for many of the cases. The authors mentioned, "we measure the detection performance when we train ResNet by varying the number of training data and assigning random label to training data on CIFAR-10 dataset." This is not exactly a fair comparison between the proposed method to the others since the other two methods are based on the softmax prediction. In cases, where we do not have enough training data, the classifiers will probably perform poorly and then the confidence won't be as good either. Lastly, the results are surprisingly good and improve recent SOTA by a huge gap. I would expect the authors to provide some explanation for such an improvement besides visualizing of the outputs using t-sne (which is kind of obvious). Minor comments: Line 171: "In the case, we expect" -> "In that case..." Line 267: "we keep 5 exemplars" -> "we keep 5 exempels" Line 227: "... and assigning random label to training data on CIFAR-10 dataset." - > "..and assigning random labels to training data on CIFAR-10 dataset." ---- RESPONSE ---- I would like to thanks the authors for putting in the effort in their response. The authors made some things clearer for me in their response, but I still think they could do a better job performing controlled experiments and properly evaluate the different components of their method. Esspatioaly after gaining such massive inprovment over previous SOTA.